# Glaucoma Cases Following SARS-CoV-2 Vaccination: A VAERS Database Analysis

**DOI:** 10.3390/vaccines10101630

**Published:** 2022-09-28

**Authors:** Rohan Bir Singh, Uday Pratap Singh Parmar, Wonkyung Cho, Parul Ichhpujani

**Affiliations:** 1Massachusetts Eye and Ear, Department of Ophthalmology, Harvard Medical School, Boston, MA 02114, USA; 2Department of Ophthalmology, Leiden University Medical Center, 2333 Leiden, The Netherlands; 3Discipline of Ophthalmology and Visual Sciences, Faculty of Health and Medical Sciences, Adelaide Medical School, University of Adelaide, Adelaide 5000, Australia; 4Glaucoma Service, Department of Ophthalmology, Government Medical College and Hospital, Chandigarh 160019, India

**Keywords:** SARS-CoV-2 vaccines, glaucoma, VAERS, adverse events, COVID-19

## Abstract

**Background:** To counter the rapidly spreading severe acute respiratory syndrome coronavirus 2 (SARS-CoV-2), global vaccination efforts were initiated in December 2020. We assess the risk of glaucoma following SARS-CoV-2 vaccination and evaluate its onset interval and clinical presentations in patients. **Methods:** We performed a retrospective analysis of the glaucoma cases reported to the Vaccine Adverse Event Reporting System (VAERS) database between 16 December 2020, and 30 April 2022. We assessed the crude reporting rate of glaucoma, clinical presentations, onset duration, and associated risk factors. **Results:** During this period, 161 glaucoma cases were reported, with crude reporting rates (per million doses) of 0.09, 0.06, and 0.07 for BNT162b2, mRNA-1273, and Ad26.COV2.S, respectively. The mean age of the patients was 60.41 ± 17.56 years, and 67.7% were women. More than half (56.6%) of the cases were reported within the first week of vaccination. The cumulative-incidence analysis showed a higher risk of glaucoma in patients who received the BNT162b2 vaccines compared with mRNA-1273 (*p* = 0.05). **Conclusions:** The incidence of glaucoma following vaccination with BNT162b2, mRNA-1273, or Ad26.COV2.S is extremely rare. Amongst the patients diagnosed with glaucoma, the onset interval of adverse events was shorter among those who received the BNT162b2 and rAd26.COV2.S vaccines compared with mRNA-1273. Most glaucoma cases were reported within the first week following vaccination in female patients and from the fifth to seventh decade. This study provides insights into the possible temporal association between reported glaucoma events and SARS-CoV-2 vaccines; however, further investigations are required to identify the potential causality link and pathological mechanisms.

## 1. Introduction

In response to the COVID-19 pandemic, large-scale global vaccination efforts were launched in December 2020 [1]. As the COVID-19 vaccines were approved for emergency-use authorization by the United States Food and Drug Administration (FDA), the Centers for Disease Control and Prevention (CDC) expanded its passive surveillance system (known as the Vaccine Adverse Event Reporting System (VAERS)) to include a wide array of adverse events of interest, including ophthalmic disorders, such as glaucoma [2].

Glaucoma is a group of ocular disorders that cause characteristic optic neuropathy with corresponding visual-field defects that result from the progressive degeneration of the retinal ganglion cells in the optic disc and the loss of their axons in the optic nerve [3]. In 2020, an estimated 76.02 million people were reported to have glaucoma worldwide [4]. A raised intraocular pressure (IOP) has been identified as a significant risk factor. These pathological changes lead to progressive peripheral-visual-field defects and may result in blindness. Despite the clinical insights into primary and secondary glaucoma, the pathological mechanisms at the cellular and subcellular levels are poorly understood, and the factors that significantly contribute to the disease progression are yet to be delineated [5]. Although several reports highlight ocular adverse events associated with COVID-19 vaccination, the literature associating COVID-19 vaccination with glaucoma is sparse. Moreover, the COVID-19 vaccine-related information portals categorically report the absence of evidence linking vaccination with glaucoma.

We used the VAERS database to evaluate the reports of glaucoma cases for three FDA-approved COVID-19 vaccines: BNT162b2 (Pfizer Inc./BioNTech SE, Mainz, Germany), mRNA-1273 (Moderna Therapeutics Inc., Cambridge, MA, USA), and Ad26.COV2.S (Janssen Pharmaceuticals, Beerse, Belgium). We determined the crude reporting rate of glaucoma in vaccine recipients since the initiation of the vaccination program. Additionally, we assessed the clinical characteristics and association between age, sex, and duration of onset (following vaccination) in patients who received one of the three vaccines.

## 2. Methods

VAERS is a passive surveillance platform that functions as an early warning system for potential vaccine related adverse events [2]. The VAERS data are available through Wide-ranging Online Data for Epidemiologic Research (WONDER), a database developed and operated by the CDC, an agency of the United States federal government. The VAERS database compiles reports of all post-vaccination adverse events from patients, parents (for minor patients), clinicians, vaccine manufacturers, and regulatory bodies globally. The database includes a detailed report of the adverse events experienced by patients following vaccination. The data recorded in VAERS are verified by third-party professional coders who assign appropriate medical terminology (based on the preferred terms of the Medical Dictionary for Regulatory Activities) from the data in the submitted reports [6]. A false VAERS report violates federal law (18 U.S. Code § 1001) and is punishable by a fine and imprisonment. CDC WONDER allows access to the information freely, as well as the use, copying, distributing, or publishing of this information without additional or explicit permission [7]. This study was conducted in compliance with the tenets of the Declaration of Helsinki and the National Statement on Ethical Conduct in Human Research 2007. As the study includes publicly available, deidentified anonymous data, the University of Adelaide Human Research Ethics Committee exempted it from ethical review.

The VAERS data included in this study were accessed via CDC WONDER on April 30, 2022 [8]. The data query included all SARS-CoV-2 vaccine-adverse events for all vaccine types administered to patients of all ages and genders for glaucoma (unspecified type), angle-closure glaucoma, open-angle glaucoma, uveitic glaucoma, and uveitis-glaucoma–hyphema syndrome. The results were grouped by symptoms, age, sex, state/territory, and onset interval. The additional measures included in the results were—an adverse-event description, lab data, current illness, adverse events after prior vaccination, medications at the time of vaccination, and history/allergies. The data included in the analysis were clinical presentation, date of vaccination and adverse-event onset, ocular and systemic history, and prescribed drugs and surgeries performed on the patients before the presentation. Some of the patients’ reports also included the interventions post-glaucoma diagnosis.

The unspecified glaucoma data were stratified broadly into open-angle glaucoma (OAG) and angle-closure glaucoma (ACG), as per the reported clinical presentations by a glaucoma specialist (PI). The patients presenting with elevated IOP and general ocular symptoms, such as eye pain without any associated ocular morbidity, were broadly considered open-angle glaucoma. The patients who were categorically reported with vision loss/blindness associated with colored haloes and redness were classified as angle-closure glaucoma. However, this classification may have discrepancies as the gonioscopy findings were not reported.

## 3. Statistical Analysis

The statistical analysis was performed using R Studio (R Foundation for Statistical Computing, Vienna, Austria). The crude reporting rates were estimated using the number of glaucoma reports (by vaccine type) per million SARS-CoV-2 vaccine doses administered. We performed a descriptive analysis of the social and demographic characteristics and vaccination data. We assessed the association between the onset interval of glaucoma and the vaccine type, age, sex, and dosage using the one-way analysis of variance (ANOVA) test. The *t*-test was used to evaluate the association between glaucoma diagnosis and a prior history of COVID-19. A cumulative-incidence analysis was performed for BNT162b2 and mRNA-1273 vaccines. The Ad26.COV2.S vaccine was excluded from this analysis due to few reports. As the primary outcome measure (i.e., glaucoma diagnosis) was categorical, the analysis was performed to investigate the risk factors associated with it using Pearson’s chi-square test of association. The missing values in the data were indicated and Na.rm code was used to account for them during the analysis. The onset-interval data are reported as means ± standard errors of means (SEMs).

## 4. Results

A total of 2,061,557,270 COVID-19 vaccine doses were administered during the study period: 80.7% were BNT162b2, 16.8% were mRNA-1273, and 2.5% were Ad26.COV2.S [1]. During this period, 1,250,310 (0.06% of all doses) adverse events post-COVID-19 vaccinations were recorded in CDC VAERS, including 166 reports of glaucoma [2]. In our analysis, 161 reports were included, four were duplicated, and one report did not include any information except the type of vaccine administered. The cases were reported by drug regulatory agencies (*n* = 99, 61.5%), physicians (*n* = 18, 11.2%), directly by patients (*n* = 26, 16.1%), and vaccine manufacturers (*n* = 16, 9.9%). The estimated crude reporting rates (per million doses) for BNT162b2, mRNA-1273, and Ad26.COV2.S were 0.09, 0.06, and 0.07, respectively. All the cases in the cohort were classified as “medically significant” adverse events by the CDC VAERS. The average age of the patients included in the study was 60.41 ± 17.56 years, and 67.7% (*n* = 109) were female (Table 1). The cases were reported from the United States (48, 28.81%), Europe (86, 53.4%), and Asia (18, 11.2%) (Appendix A). The state-by-state crude reporting rates for the three vaccines administered in the United States are reported in Appendix A.

In the study cohort, the majority of the patients (*n* = 130, 80.7%) were administered the BNT162b2 vaccine, while 27 patients (16.8%) received the mRNA-1273 vaccine, and four patients (2.5%) received the rAd26.COV2.S vaccine (Table 2). Most of the cases (*n* = 91, 56.5%) were reported within the first week, including 18% (*n* = 29) of the cases on the day of vaccination (Figure 1).

A total of 77 cases (47.8%) were reported after the first dose, 59 cases (36.6%) were reported after the second dose, and 13 cases (8.1%) were reported after the third dose of the vaccine (Figure 2). Only five patients (3.1%) reported a prior history of COVID-19. On stratifying the patients based on the clinical descriptions, we found that most of the cases (*n* = 105, 65.2%) had OAG. The patients presented with ocular pain (*n* = 60, 37.3%), reduced/blurry vision (47, 29.2%), and complete vision loss/blindness (*n* = 35, 21.7%). An elevated IOP was reported in 48 (29.8%) cases. The other ocular signs included flashes (*n* = 6, 3.7%), floaters (*n* = 5, 3.1%), and photophobia (*n* = 5, 3.1%). Notably, 28 patients (17.3%) had a prior history of glaucoma and had controlled IOP at the time of vaccination. Seven patients (4.3%) had a history of uveitis. The patients also presented with severe headache (*n* = 48, 29.8%), general body pain (*n* = 17, 10.6%), and high blood pressure (*n* = 8, 5.0%). The patients had a prior history of cardiovascular diseases (*n* = 33, 20.5%) and hypertension (*n* = 13, 8.1%). The ocular and systemic presentation and history of the patients included in the study are summarized in Table 3.

The statistical evaluation revealed a significant association between the vaccine type and the glaucoma-onset duration following vaccination. The symptom onset duration in the patients who received the BNT162b2 (14.7 ± 27.58 days) and Ad26.COV2.S (5.5 ± 6.4 days) vaccines were significantly shorter compared with those who received mRNA-1273 (37.07 ± 66.11 days, *p* = 0.013) (Table 4). These findings were confirmed by a cumulative-incidence analysis, which showed a significant difference in the onset duration between BNT162b2 and mRNA-1273 (*p* = 0.05) (Figure 3). The analysis showed that there was no significant association between the onset interval and sex (*p* = 0.196), age (0.565), and history of COVID-19 (*p* = 0.08). Pearson’s chi-square analysis showed that the frequency of glaucoma cases in patients who received the mRNA-1273 vaccine was significantly higher (*p* = 0.047) in the older age groups (sixth–seventh decade) (Table 5). A similar association was not observed in patients vaccinated with the BNT162b2 and Ad26.COV2.S vaccines. Additionally, we did not observe any significant association between the vaccine dose, sex, and onset interval.

The patients were prescribed topical eye drops (*n* = 44), and surgical interventions (*n* = 29) were performed in the requisite cases. Laser iridotomy and shunt/valve placement surgeries were performed in 18 (11.2%) and 4 (2.5%) patients, respectively. None of the patients included in this study had a pre-existing iridotomy. Trabeculectomies were performed in two (1.2%) patients. The type of surgical intervention was not specified in seven (4.3%) patients. The patients were prescribed brimonidine (*n* = 5, 3.1%), dorzolamide (*n* = 6, 3.7%), timolol (*n* = 11, 6.8%), or Travoprost/Latanoprost (*n* = 2, 1.2%). The prescribed eye drops were not specified in 18 (11.2%) patients. The interventions were not known for more than half (*n* = 86, 53.4%) of the patients (Appendix A).

## 5. Discussion

Since initiation of the global vaccination in December 2020, several case reports have highlighted the ocular adverse events associated with the COVID-19 vaccines. The current study evaluates the temporal association between glaucoma and the SARS-CoV-2 vaccines, as there was a short interval between post-vaccination and the onset of the signs of glaucoma; however, further studies are required to evaluate the potential causal relationship [9]. Over the years, several studies have established the presence of the renin-angiotensin system in the human ciliary body and aqueous humor [10,11,12]. It has been speculated that binding of the spike proteins generated by SARS-CoV-2 vaccines (such as BNT162b2 and mRNA-1273) to angiotensin-converting enzyme 2 (ACE2) leads to angiotensin II overactivity, thereby increasing the aqueous-humor production and elevated intraocular pressure [13].

The ocular adverse events associated with the three vaccines approved in the United States include uveitis, Bell’s and abducens nerve palsy, acute macular neuroretinopathy, central serous retinopathy, Grave’s disease, Vogt–Koyanagi–Harada disease, retinal and ophthalmic vein thrombosis, and corneal-graft rejection [14,15,16,17,18,19,20,21,22,23]. Similar ocular disorders have also been observed in patients infected by the SARS-CoV-2 virus [24,25,26]. Recently, Choi and colleagues reported vision-threatening ocular adverse events in sixteen patients following SARS-CoV-2 vaccination [27]. Among these patients, the authors reported four cases of angle-closure glaucoma following the ChAdOx1 nCoV-19 vaccine and attributed it to the inflammation of the ciliary body due to vaccine-associated uveitis. Additionally, two cases of secondary angle-closure glaucoma post-vaccination have been reported in the literature. Behera and colleagues reported the case of a 60-year-old male with hemophilia who developed painful and sudden vision loss a day after receiving the ChAdOx1 nCoV-19 (Oxford AstraZeneca) vaccine due to acute angle-closure glaucoma secondary to a massive suprachoroidal hemorrhage [28]. In the other case, a 49-year-old male presented with progressive vision loss one day after the administration of the second dose of the BNT162b2 vaccine. The patient had a massive intraocular hemorrhage and was diagnosed with secondary angle-closure glaucoma, bullous retinal detachment, and massive intraocular hemorrhage [29]. The patient’s presentation was attributed to the necrosis of a melanocytic lesion at the posterior edge of the ciliary body and choroid [29]. In another case report, Santovito and Pinna reported reduced vision, severe headache, and photophobia in a patient after vaccination with BNT162b2. The patient had no prior ocular history, and the authors could not establish a definitive diagnosis for the patient [16]. The pathogenesis of secondary glaucoma following vaccination can be explained by the underlying disorders, as observed in the abovementioned cases, or by trabecular dysfunction that is likely mediated by inflammatory responses and oxidative stress. Additionally, the fluctuations in the intraocular pressure also play a critical role in inducing metabolic stress [3].

The analysis of the VAERS data suggests an extremely low safety concern for glaucoma post-vaccination with BNT162b2, mRNA-1273, and Ad26.COV2.S. The estimated crude reporting rates in this study are comparable with the report by Wang and colleagues, who evaluated the data from the Australia Therapeutic Goods Administration Database of Adverse Event Notifications, the Canada Vigilance Adverse Reaction Database, the European Union Medicines Agency (EudraVigilance) System, and the United Kingdom Medicines and Healthcare Products Regulatory Agency between December 2020 and August 2021 [30]. Although the glaucoma crude reporting rate is extremely low, it may be considered a “severe” adverse event because the fluctuations in the intraocular pressure can have a domino effect on the aqueous dynamics and optic nerve.

In this study, a substantial proportion of the patients were women and between 50 and 70 years old. The data analysis of this cohort shows that the patients typically presented with signs of glaucoma in the first 24–48 h post-vaccination (viz., elevated intraocular pressure), and the incidence was higher after the first dose. Most of the patients did not have a prior history of glaucoma; therefore, it is imperative that patients at risk of developing glaucoma remain vigilant post-vaccination. The onset interval of the disease was significantly shorter in the patients vaccinated with the BNT162b2 and Ad26.COV2.S vaccines compared with those vaccinated with the mRNA-1273 vaccine. The patients diagnosed with glaucoma after mRNA-1273 vaccination were between 60 and 80 years old. Although uveitis and other ocular inflammatory disorders are considered more common ocular adverse events associated with vaccination, only 4% of the patients were diagnosed with it. The pathogenesis of secondary glaucoma post-vaccination can be explained by the underlying disorders or trabecular dysfunction likely mediated by inflammatory responses and oxidative stress. In addition, increases or fluctuations in the intraocular pressure induce metabolic stress. However, there are no insights into the pathophysiologic mechanisms that can potentially cause open-angle glaucoma reported in many of the patients included in this study.

## 6. Limitations

This study, reporting glaucoma cases following SARS-CoV-2 vaccination, has several limitations. VAERS is a passive surveillance system which includes adverse event reports following administration of FDA approved vaccines, from pharmaceutical companies, physicians, drug regulators, and patients from all over the world. Despite the mandatory requirement to report vaccine-associated adverse events, underreporting and delayed reporting are common. In some cases, the submitted reports are incomplete and lack uniformity in the data reporting. Several reports have missing data points, such as ethnicity, that are considered important risk factors associated with glaucoma.

The absence of an unvaccinated control group impedes the relative-risk calculation. The pharmacovigilance associated with COVID-19 vaccines is limited to the European Union, the United States, Australia, Canada, and a few Asian countries. Hence, reports are not recorded from many developing countries, including India, where over one billion vaccine doses have been administered. Moreover, the data are absent for other approved vaccines, such as ChAdOx1 nCoV-19, ZyCoV-D, Sputnik, Covidecia, Sputnik, Sinopharm, Abdala, Zifivax, and Novavax.

The reports submitted by drug regulators, pharmaceutical companies, and physicians (~80%) can be relied on for the clinical diagnosis of glaucoma; however, cases in which patients self-reported are presumed to be glaucoma. The data in the literature suggest glaucoma underreporting; therefore, it can be assumed that the cases were grossly underreported. The lack of clinical (specifically vision and gonioscopy) data severely impedes the use of these data on the ability of the researchers to report the adverse effects in these reports convincingly. Additionally, there are no details of optic-nerve head cupping or visual-field defects on the perimetries; thus, what is labeled as glaucoma could also be secondary ocular hypertension, as access to detailed clinical history is not logistically possible and is limited by the Health Insurance Portability and Accountability Act (HIPPA) of 1996 [31].

In conclusion, the risk of glaucoma following vaccination with BNT162b2, mRNA-1273, and rAd26.COV2.S is extremely low. The majority of the patients in the cohort had primary open-angle glaucoma and had received the BNT162b2 vaccine. Most cases occurred after the first dose and within the first week following vaccination. Therefore, it is recommended that ophthalmologists and glaucoma specialists closely monitor the at-risk patients following vaccination.

## Figures and Tables

**Figure 1 vaccines-10-01630-f001:**
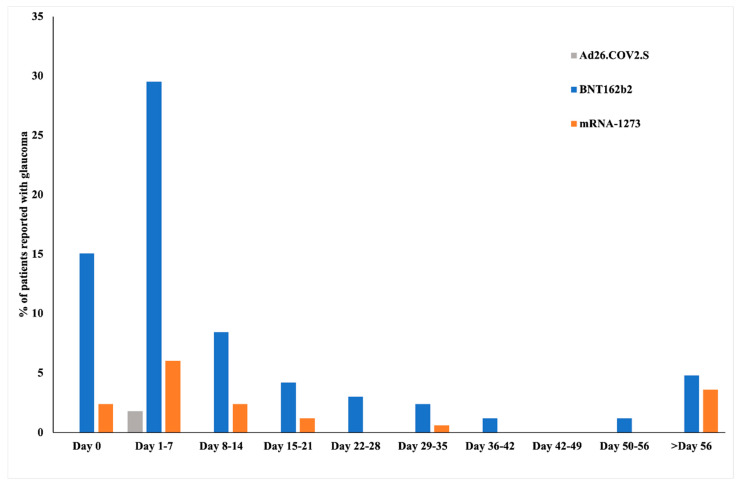
Cases of glaucoma following vaccination with BNT162b2, mRNA-1273, and Ad26.COV2.S on day 0 (i.e., day of vaccination) and in subsequent weeks.

**Figure 2 vaccines-10-01630-f002:**
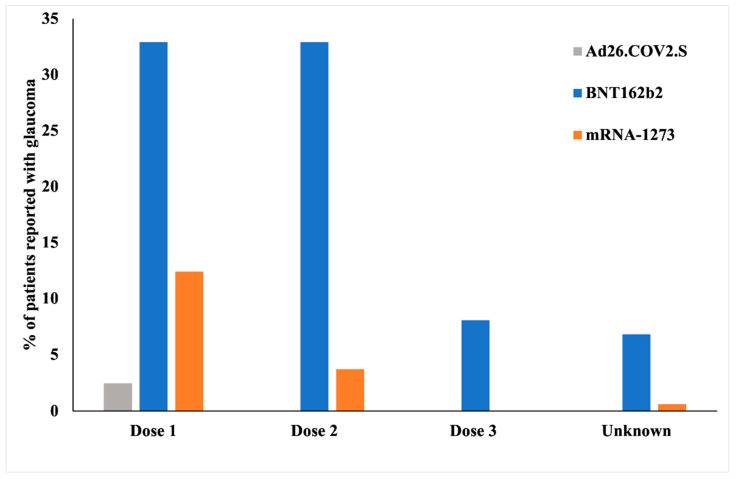
Cases of glaucoma post-vaccination with protocol doses (Doses 1 and 2 for BNT162b2 and mRNA-1273 and Dose 1 for Ad26.COV2.S) and boosters.

**Figure 3 vaccines-10-01630-f003:**
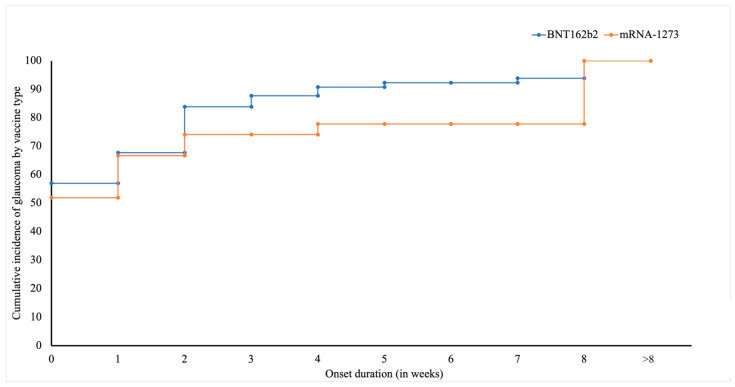
Cumulative-incidence analysis for glaucoma cases reported after administration of BNT162b2 and mRNA-1273 vaccines.

**Table 1 vaccines-10-01630-t001:** Demographics of patients who were diagnosed with glaucoma following SARS-CoV-2 vaccination.

	Frequency (*n*)	%
**Mean Age (in years)**	60.41 ± 17.56	
**Age Range**		
10–19	2	1.2
20–29	9	5.6
30–39	9	5.6
40–49	9	5.6
50–59	42	26.1
60–69	21	13.0
70–79	38	23.6
80–89	16	9.9
90+	4	2.5
Unknown	11	6.8
**Sex**		
Female	109	67.7
Male	51	31.7
Unknown	1	0.6
**Origin**		
Australia	1	0.6
Asia	18	11.2
Europe	86	53.4
United States	48	29.8
Foreign (nonspecific)	3	1.9
Unknown	5	3.1

**Table 2 vaccines-10-01630-t002:** Vaccination data of patients who were diagnosed with glaucoma following vaccination.

	Frequency (*n*)	%
**Type of Vaccine**		
Ad26.COV2.S	4	2.5
BNT162b2	130	80.7
mRNA-1273	27	16.8
**Dosage**		
1	77	47.8
2	59	36.6
3	13	8.1
Unknown	12	7.5
**Median Onset Interval** **(in days)**	4	
**Onset Interval Post-Vaccination**		
Day 0	29	18.0
Days 1–7	62	38.5
Days 8–14	18	11.2
Days 15–21	9	5.6
Days 22–28	5	3.1
Days 29–35	5	3.1
Days 36–42	2	1.2
Days 42–49	0	0.0
Days 50–56	2	1.2
Days 56+	14	8.7
Unknown	15	9.3

**Table 3 vaccines-10-01630-t003:** Ocular and systemic history and presentation in patients diagnosed with glaucoma following SARS-CoV-2 vaccination.

	Frequency	%
**History of COVID-19**	5	3.1
**Glaucoma Diagnosis**		
Open-angle glaucoma	105	65.2
Angle-closure glaucoma	45	27.6
Unknown	11	6.8
**Ocular Presentations**		
Flashes	6	3.7
Floaters	5	3.1
High IOP	48	29.8
Ocular pain	60	37.3
Photophobia	5	3.1
Reduced/blurry vision	47	29.2
Vision loss/blindness	35	21.7
**Ocular History**		
Conjunctivitis	4	2.5
Glaucoma (controlled)	28	17.3
Hemorrhage	3	1.9
Herpes zoster ophthalmicus	5	3.1
Keratitis	2	1.2
Ocular ischemic syndrome	1	0.6
Optic ischemic neuropathy	1	0.6
Retinal artery occlusion	1	0.6
Retinal vein occlusion	6	3.7
Uveitis	7	4.3
Vitreous detachment	2	1.2
**Systemic Presentation**		
Headache	48	29.8
Pain	17	10.6
Nausea	12	7.5
Palpitations	4	2.5
High blood pressure	8	5.0
Vaccine site induration/rash	6	3.7
**Systemic History**		
Allergies	9	5.6
Cardiovascular disorders (MI, CAD, Afib, tachyarrhythmia, heart failure)	33	20.5
Cerebrovascular disorders	3	1.9
Diabetes	9	5.6
Hypercholesterolemia/Dyslipidemia	7	4.3
Hypertension	13	8.1
Hypothyroidism	11	6.8
Pulmonary disorders (COPD, embolism)	4	2.5
Renal disorders	5	3.1

**Table 4 vaccines-10-01630-t004:** Analysis to assess factors associated with onset interval of glaucoma following SARS-CoV-2 vaccination.

	Percentage (*n*)	Mean Onset Interval (in Days)	Median Onset Interval (in Days)	*p*-Value
**Vaccine ***				
BNT162b2	80.7% (130/161)	14.7 ± 2.42	5	**0.013 ***
mRNA-1273	16.7% (27/161)	37.07 ± 12.71	7	
Ad26.COV2.S	2.5% (4/161)	5.5 ± 3.2	3	
**Sex ***				
Female	67.7% (109/161)	15.17 ± 2.98	4	0.196
Male	31.7% (51/161)	24.6 ± 6.65	9	
**Age ***				
10–19	1.86% (3/161)	32 ± 31.03	1	0.565
20–29	5.6% (9/161)	21.88 ± 17.74	1	
30–39	6.8% (11/161)	29.81 ± 23.54	5	
40–49	8% (13/161)	12.46 ± 4.95	5	
50–59	32.3% (52/161)	14.90 ± 3.65	3.5	
60–69	13% (21/161)	27.04 ± 8.62	8	
70–79	21.7% (35/161)	15.57 ± 5.23	6	
80–89	8.7% (14/161)	20.71 ± 12.13	10	
90+	1.86% (3/161)	12 ± 10.51	2	
**Dosage ***				
1	47.8% (77/161)	15.34 ± 4.26	4	0.268
2	36.6% (59/161)	23.64 ± 5.60	9	
3	8.07% (13/161)	17.61 ± 6.35	6	
Unknown	7.45% (12/161)			
**History of COVID-19 ****				
Yes	3.1% (5/161)	54.2 ± 32.5	33	0.08
No	96.27% (155/161)	17.18 ± 2.86	5	
Unknown	0.62% (1/161)			

* One-way ANOVA test; ** *t*-test.

**Table 5 vaccines-10-01630-t005:** Association analysis of age, sex, and onset interval with glaucoma following SARS-CoV-2 vaccination.

	BNT162b2 (Pfizer BioNTech)						mRNA-1273					Ad26.COV2.S			
	Unknown	1	2	3	*χ* ^2^	*p*-value	Unknown	1	2	*χ* ^2^	*p*-value	Unknown	1	*χ* ^2^	*p*-value
**Age (in years)**					19.17	0.742				26.2	**0.047 ***			1.333	0.995
10–19	0	0	2	0			0	0	0			0	1		
20–29	0	2	4	0			0	1	0			0	0		
30–39	2	6	2	0			1	1	0			0	0		
40–49	1	5	4	1			0	1	0			0	0		
50–59	4	16	16	4			0	0	1			0	2		
60–69	0	4	7	3			0	9	1			0	1		
70–79	1	14	13	4			0	5	2			0	0		
80–89	3	5	4	1			0	3	0			0	0		
90+	0	1	1	0			0	0	2			0	0		
**Sex**					0.555	0.906				0.905	0.636			1.333	0.248
Male	3	16	15	5			0	8	3			0	2		
Female	8	37	38	8			1	12	3			0	2		
**Onset Interval (in days)**					21.08	0.275				16.78	0.157			1.333	0.987
0	1	9	13	2			0	4	0			0	0		
1–7	3	25	16	5			1	9	0			0	3		
8–14	1	4	8	1			0	2	2			0	0		
15–21	4	10	6	1			0	2	0			0	1		
22–28	1	1	2	1			0	0	0			0	0		
29–35	1	1	2	0			0	0	1			0	0		
36–42	0	1	1	0			0	0	0			0	0		
43–49	0	0	0	0			0	0	0			0	0		
49–56	0	0	0	2			0	0	0			0	0		
56+	0	2	5	1			0	3	3			0	0		

* *p* < 0.05.

## Data Availability

Publicly available datasets were analyzed in this study. These data can be found at https://wonder.cdc.gov/vaers.html (accessed on 30 April 2022).

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
