# Peer review of "Glaucoma Cases Following SARS-CoV-2 Vaccination: A VAERS Database Analysis"

_vaccines, 2022, doi:10.3390/vaccines10101630_

Round 1

Reviewer 1 Report (Previous Reviewer 2)

1.         The reverse Kaplan-Meier analysis is for censoring analysis, which reverse the censor time and the event time, not for survival probability and one minus survival probability. Due to no censoring data in this n=161 cohort, the results will be the same. But the major interesting is the onset time to glaucoma event, using Kaplan-Meier estimator is appropriated.

2.         Figure 3: The cumulative incidence curve should be a step function, need to be corrected.

Author Response

Dear Prof. Tripp

We thank you and the reviewers for the valuable feedback and for allowing us to resubmit the manuscript for consideration for publication in Vaccines. We have carefully considered all of the comments and revised the manuscript accordingly. The suggested revisions have helped in improving this manuscript. We have addressed each of the comments below in a point-by-point response below:

REVIEWER 1

  1. The reverse Kaplan-Meier analysis is for censoring analysis, which reverse the censor time and the event time, not for survival probability and one minus survival probability. Due to no censoring data in this n=161 cohort, the results will be the same. But the major interesting is the onset time to glaucoma event, using Kaplan-Meier estimator is appropriated.

Response: We thank the reviewer for their constructive feedback. We concur with the reviewer’s viewpoint about the inappropriate use of reverse Kaplan-Meier analysis. As per the reviewer’s feedback we have changed it to cumulative incidence analysis. As the reviewer rightly pointed out the results remain unchanged.

  1. Figure 3: The cumulative incidence curve should be a step function, need to be corrected.

Response: The requisite changes have been made as per the reviewer’s feedback.

Reviewer 2 Report (New Reviewer)

This is a valuable epidemiological study, performed a descriptive analysis of the social demographic characteristics and vaccination data to present ocular adverse effect post-COVID-19 vaccination. The authors using a very valuable database, but with imperfect and incomplete report data, successfully pointed out the higher risk of glaucoma in BNT1612b2 vaccine. However, I still have some minor question and suggestion as the following.

1.Please add more detail discussion on the different pathophysiologic mechanisms of various types of glaucoma (glaucoma unspecified type, angle-closure glaucoma, open-angle glaucoma, uveitis glaucoma, uveitis-glaucoma-hyphaema syndrome etc.) and different action mechanism of vaccines (BNT162b2, mRNA-1273, or Ad26.COV2. S.)? And please discuss about the potential biological plausibility between BNT162b2 and glaucoma.
2. The author mention that “The absence of an unvaccinated control group impedes relative risk calculation”. Is there any possibility by other study design, such as case-control study, could provide more explanation about the result?

Author Response

Dear Prof. Tripp

We thank you and the reviewers for the valuable feedback and for allowing us to resubmit the manuscript for consideration for publication in Vaccines. We have carefully considered all of the comments and revised the manuscript accordingly. The suggested revisions have helped in improving this manuscript. We have addressed each of the comments below in a point-by-point response (in italics) below:

REVIEWER 2

  1. This is a valuable epidemiological study, performed a descriptive analysis of the social demographic characteristics and vaccination data to present ocular adverse effect post-COVID-19 vaccination. The authors using a very valuable database, but with imperfect and incomplete report data, successfully pointed out the higher risk of glaucoma in BNT1612b2 vaccine. However, I still have some minor question and suggestion as the following.

Response: We thank the reviewer for their positive feedback. We agree with the reviewer regarding the shortcomings of the VAERS database.

1.Please add more detail discussion on the different pathophysiologic mechanisms of various types of glaucoma (glaucoma unspecified type, angle-closure glaucoma, open-angle glaucoma, uveitis glaucoma, uveitis-glaucoma-hyphaema syndrome etc.) and different action mechanism of vaccines (BNT162b2, mRNA-1273, or Ad26.COV2. S.)? And please discuss about the potential biological plausibility between BNT162b2 and glaucoma.

Response: As per the reviewer’s recommendation we have added brief details about the possible underlying pathological mechanisms that cause glaucoma following SARS-CoV-2 vaccine. We have intentionally tried to not speculate about the immunological mechanisms due to the lack of studies that establish the causal relationship between the vaccines and glaucoma onset. It has been speculated that the binding of spike proteins generated by SARS-CoV-2 vaccines (such as BNT162b2 and mRNA-1273) with angiotensin converting enzyme 2 (ACE2) may lead to angiotensin II overactivity, thereby increasing aqueous humor production and elevated intraocular pressure. We have added these details to the discussion section.

  1. The author mention that “The absence of an unvaccinated control group impedes relative risk calculation”. Is there any possibility by other study design, such as case-control study, could provide more explanation about the result?

Response: We thank the reviewer for this suggestion. We do plan to conduct a multi-centric case control study to evaluate the risk of various ocular disorders following SARS-CoV-2 vaccination. As this study was primarily focused on the data reported to CDC-VAERS database, we tried to adopt the best possible approach for this analysis considering all the limitations.

Round 2

Reviewer 1 Report (Previous Reviewer 2)

Figure 3 should be renewed. The cumulative incidence curve is a step function, like stairs, as following picture.

Author Response

Figure 3 should be renewed. The cumulative incidence curve is a step function, like stairs, as following picture.

Response: The figure has been updated as step-function as per the reviewer's feedback. We apologize for not understanding the initial comment from the reviewer. 

Round 3

Reviewer 1 Report (Previous Reviewer 2)

Figure 3 is acceptable. 

Author Response

We thank the reviewer for their positive feedback.

This manuscript is a resubmission of an earlier submission. The following is a list of the peer review reports and author responses from that submission.

Round 1

Reviewer 1 Report

The authors present the analysis of the association between glaucoma and prior history of COVID-19 with the help of statistical analysis on VAERS data. The authors focused on 200 glaucoma cases out of ~1.2 million reports from VAERS. It is interesting to notice that the authors mentioned the limitations of VAERS data (as it is public and does not undergo quality assurance standards and may pose challenges when statistical analysis approaches are applied). Additionally, the authors mentioned the issues of duplicates in the dataset which can pose deleterious effects on machine learning/statistical analysis of VAERS data. However, the authors should highlight the point that only 200 glaucoma cases across 1.2 million covid vaccines reports may not be significant.    Have the conducted any text mining of the "patient report" field, or if they used the symptoms fields from VAERS? If they used the patient report field, it is important to ensure that the word "glaucoma" is being used in the correct way (which the authors could have done manually since there are only 200 glaucoma reports). For example, "the covid vaccine has given me glaucoma" vs. "the covid vaccine is making my glaucoma flare up" are two different perspectives/facts. On the other hand, if the symptom field was used, it necessarily implies that the glaucoma cases are presented.   Below are some minor comments:   Abstract:  The abstract mentions that 67.7% of the patients presenting with glaucoma were women. While this is true, it might be a misinterpretation given that ~66% of the VAERS reports represent female in the first place.   The authors report "crude reporting of glaucoma". What do the authors mean by crude reporting specifically?   "The mean age of patients was 60.41±17.56 years, and 67.7% were women". It is important to note that the VAERS dataset in its entirety provides a bias between gender with 70% women. It could be misinterpreted easily   "More than half (56.6%) were reported within the first week after vaccination". It might be interesting to know the reporting frequency of the symptoms with respect to time (e.g., how many symptoms in the first week vs the remaining period).   Introduction:   Lines 79 - 82: Study referenced in [8]. Please specify what column in VAERS was the study focused on. Was this done for the detailed report column or the "appropriate medical terminology" column? Or both?   Line 134: It would be interesting to compare these statistics against the overall percentages. What percentage of VAERS was discovered for Pfizer vs the others?   Discussion:    The authors report "In this study cohort, a substantial proportion of the patients were women and be tween 50-70 years". It is again important to note how these groups relate to the entirety of VAERS   Lastly, a recent study (copied below) is published highlighting important issues to consider while carrying out the statistical analysis on VAERS data. The authors are encouraged to review and compare the protocols followed in their study.   Flora, J.; Khan, W.; Jin, J.; Jin, D.; Hussain, A.; Dajani, K.; Khan, B. Usefulness of Vaccine Adverse Event Reporting System for Machine-Learning Based Vaccine Research: A Case Study for COVID-19 Vaccines. Int. J. Mol. Sci. 2022, 23, 8235. https://doi.org/10.3390/ijms23158235

Author Response

Dear Prof. Tripp

We thank you and the reviewer for the valuable feedback and for allowing us to resubmit the manuscript for consideration for publication in Vaccines. We have carefully considered all of the comments and revised the manuscript accordingly. The suggested revisions have helped in improving this manuscript. We have addressed each of the comments below in a point-by-point response (in italics) below:

REVIEWER 1

  1. The authors present the analysis of the association between glaucoma and prior history of COVID-19 with the help of statistical analysis on VAERS data. The authors focused on 200 glaucoma cases out of ~1.2 million reports from VAERS. It is interesting to notice that the authors mentioned the limitations of VAERS data (as it is public and does not undergo quality assurance standards and may pose challenges when statistical analysis approaches are applied). Additionally, the authors mentioned the issues of duplicates in the dataset which can pose deleterious effects on machine learning/statistical analysis of VAERS data. However, the authors should highlight the point that only 200 glaucoma cases across 1.2 million covid vaccines reports may not be significant.

Response: We thank the reviewer for their feedback. The data included in the VAERS database does undergo quality check by CDC and submitting falsified reports to the database is punishable by fine and imprisonment under 18 U.S. Code § 1001.

We did not state at any point in the manuscript (or limitations) that there was a lack of quality assurance or duplicate reports in the database. Each case included in this study was unique. On the contrary, we feel that the cases may be grossly underreported.

We would like to bring to the reviewer’s notice that we have repeatedly highlighted it in the manuscript that glaucoma cases following SARS-CoV-2 vaccines are rare.

  1. Have the conducted any text mining of the "patient report" field, or if they used the symptoms fields from VAERS? If they used the patient report field, it is important to ensure that the word "glaucoma" is being used in the correct way (which the authors could have done manually since there are only 200 glaucoma reports). For example, "the Covid vaccine has given me glaucoma" vs. "the covid vaccine is making my glaucoma flare up" are two different perspectives/facts. On the other hand, if the symptom field was used, it necessarily implies that the glaucoma cases are presented.

Response: Each report was thoroughly evaluated by two authors. The cases which were diagnosed as glaucoma if the report categorized as “glaucoma (unspecified type), an-gle-closure glaucoma, open-angle glaucoma, uveitic glaucoma, uveitis-glaucoma-hyphaema syndrome” on the basis of MedDRA definition and presented with elevated IOP and general ocular symptoms such as eye pain without any associated ocular morbidity were considered open-angle glaucoma. The patients who were categorically reported with vision loss/blindness associated with colored haloes and redness were classified as angle-closure glaucoma. All these findings in the reports were re-checked by an experienced glaucoma specialist.

Below are some minor comments:

Abstract:

  1. The abstract mentions that 67.7% of the patients presenting with glaucoma were women. While this is true, it might be a misinterpretation given that ~66% of the VAERS reports represent female in the first place.

Response: The CDC VAERS has not highlighted reporting bias in the dataset. We feel it would be inappropriate to make that claim unless CDC or any other reliable study clearly highlights this bias in the dataset.

  1. The authors report "crude reporting of glaucoma". What do the authors mean by crude reporting specifically?

Response: The crude reporting rate is the number of glaucoma cases reported for every million doses administered. The definition has been clearly outlined in the statistical analysis section.

  1. "The mean age of patients was 60.41±17.56 years, and 67.7% were women". It is important to note that the VAERS dataset in its entirety provides a bias between gender with 70% women. It could be misinterpreted easily

Response: The CDC VAERS has not highlighted reporting bias in the dataset. We feel it would be inappropriate to make that claim unless CDC or any other reliable data source clearly highlights this bias in the dataset.

  1. "More than half (56.6%) were reported within the first week after vaccination". It might be interesting to know the reporting frequency of the symptoms with respect to time (e.g., how many symptoms in the first week vs the remaining period).

Response: We have included the reverse Kaplan-Meier risk analysis  to evaluate the cumulative incidence of glaucoma cases over time. Adding another statistical analysis of frequency with time would essentially be a repeat of above mentioned test.

  1. Introduction: Lines 79 - 82: Study referenced in [8]. Please specify what column in VAERS was the study focused on. Was this done for the detailed report column or the "appropriate medical terminology" column? Or both?

Response: We have already detailed the search protocol in the methods section. Line 80 - The data query included all SARS-CoV-2 vaccine adverse events for all vaccine types administered to patients of all ages and genders for glaucoma (unspecified type), an-gle-closure glaucoma, open-angle glaucoma, uveitic glaucoma, uvei-tis-glaucoma-hyphaema syndrome.

  1. Line 134: It would be interesting to compare these statistics against the overall percentages. What percentage of VAERS was discovered for Pfizer vs the others?

Response: This information is already outlined in the manuscript and table 2. BNT162b2 (Pfizer) had 80.7% cases and the remaining 19.3% cases were reported in patients who received the other two vaccines. We planned to do a post-hoc analysis, but did not go through with it since the patient cohort is limited to 161 patients.

  1. Discussion: The authors report "In this study cohort, a substantial proportion of the patients were women and between 50-70 years". It is again important to note how these groups relate to the entirety of VAERS

Response: The VAERS database does not provide with information for the database in its entirety. We tried to reach out to the CDC for the same, but did not receive any response.

  1. Lastly, a recent study (copied below) is published highlighting important issues to consider while carrying out the statistical analysis on VAERS data. The authors are encouraged to review and compare the protocols followed in their study.

Flora, J.; Khan, W.; Jin, J.; Jin, D.; Hussain, A.; Dajani, K.; Khan, B. Usefulness of Vaccine Adverse Event Reporting System for Machine-Learning Based Vaccine Research: A Case Study for COVID-19 Vaccines. Int. J. Mol. Sci. 2022, 23, 8235.

Response: We thank the reviewer for recommending the article. We entirely performed this data extraction manually and did not use a machine learning algorithm for the same as the data is heterogenous and non-structured.

We would like to thank the editor and the reviewer for providing us with these constructive comments and hope that with the clarifications and revisions described herein our manuscript will now be suitable for publication.

Yours sincerely,

Parul Ichhpujani on behalf of all authors

Reviewer 2 Report

1.      This manuscript illustrates the incidence of glaucoma after COVID-19 vaccination, based on the open data of VAERS system of CDC in FDA, US. There were 161 glaucoma cases reported out of the database and treated as the target cohort of this works.

2.          Line123-124: The estimated glaucoma crude reporting rate of three vaccines are acceptable. However, the estimated glaucoma crude reporting rates by origins (Australia, Asia, Europe, US in Table 1) are expected and will be better having a column with the general population glaucoma incidence rate for comparing. (If the denominator data are available.)

3.          Line108: The reverse Kaplan-Meier analysis seems not appropriate for this work, it’s for censoring analysis. Here, the major interesting is the onset time to glaucoma event, so the usual Kaplan-Meier estimator will be sufficient. During the target cohort are just the reported event, the data were no censor part. Figure 3: The cumulative incidence curve should be a step function.

4.          Table 4: It is suggested to use the usual Kaplan-Meier estimator without censoring for evaluating the difference of median onset time of glaucoma by each variable.

5.          For discussion: Does the progression of glaucoma different with and without COVID-19 vaccination? Although the glaucoma crude reporting rate is extremely low, is it a severe adverse event?

Author Response

Dear Prof. Tripp

We thank you and the reviewer for the valuable feedback and for allowing us to resubmit the manuscript for consideration for publication in Vaccines. We have carefully considered all of the comments and revised the manuscript accordingly. The suggested revisions have helped in improving this manuscript. We have addressed each of the comments below in a point-by-point response (in italics) below:

REVIEWER 2

  1.     This manuscript illustrates the incidence of glaucoma after COVID-19 vaccination, based on the open data of VAERS system of CDC in FDA, US. There were 161 glaucoma cases reported out of the database and treated as the target cohort of this works.

Response: We thank the reviewer for their constructive feedback. It has significantly improved the manuscript.

  1. Line 123-124: The estimated glaucoma crude reporting rate of three vaccines are acceptable. However, the estimated glaucoma crude reporting rates by origins (Australia, Asia, Europe, US in Table 1) are expected and will be better having a column with the general population glaucoma incidence rate for comparing. (If the denominator data are available.)

Response: We thank the reviewer for this insightful suggestion. However, the denominator data (i.e. total number of vaccines administered) is not available with the WHO global vaccination co-ordination office or the vaccine manufacturers themselves. We were able to obtain these data for the United States, hence, we evaluated the crude reporting rate for each state. We have outlined this as a limitation in the requisite section

  1. Line108: The reverse Kaplan-Meier analysis seems not appropriate for this work, it’s for censoring analysis. Here, the major interesting is the onset time to glaucoma event, so the usual Kaplan-Meier estimator will be sufficient. During the target cohort are just the reported event, the data were no censor part. Figure 3: The cumulative incidence curve should be a step function.

Response: We thank the reviewer for seeking more clarity regarding the analysis, but the reverse Kaplan-Meier risk-analysis provides a total cumulative incidence of an event over time. The referenced article clearly outlines that “One minus the Kaplan-Meier estimate of the survival function provides an estimate of the cumulative incidence of events over time.”

We have already included the onset interval and analysed the variability in different groups.

Reference: https://www.ahajournals.org/doi/10.1161/circulationaha.115.017719

  1. Table 4: It is suggested to use the usual Kaplan-Meier estimator without censoring for evaluating the difference of median onset time of glaucoma by each variable.

Response: As per reviewer’s suggestion, we have included a median onset time interval for each variable.

  1. For discussion: Does the progression of glaucoma different with and without COVID-19 vaccination? Although the glaucoma crude reporting rate is extremely low, is it a severe adverse event?

Response: We thank the reviewer for this interesting question. The clinical presentation in the cases reported to VAERS are similar to cases reported prior to global vaccination efforts. It is indeed a severe as well as an “acute” adverse event, considering >50% of the patients presented with reduced vision or complete vision loss.  Since, VAERS database has its set of limitation, more could have been known if we had structural as well as functional tests for glaucoma diagnosis.

We would like to thank the editor and the reviewer for providing us with these constructive comments and hope that with the clarifications and revisions described herein our manuscript will now be suitable for publication.

Yours sincerely,

Parul Ichhpujani on behalf of all authors

Reviewer 3 Report

Specific comments:

Line 52: Provide a reference for this statement. 

Line 123: I have reservations about the accuracy of diagnosis when patients self report glaucoma. How do the investigators know these were true glaucoma cases?

Line 141: Same problem here: COVID-19 is often asymptomatic and thus, patients not reporting it does not mean much.

Line 146: Again, patients could have had undiagnosed preexisting glaucoma. Also, with the ones having diagnosed glaucoma, the adverse event would be better described as "glaucoma exacerbation", rather than newly developed glaucoma. 

Line 159 and on: All data have an unusually high SD (higher than the datum itself). This needs to be explained. 

Line numbers disappeared for the Discussion. 

The authors provide no possible explanation why the onset was longer after the Moderna vaccine, compared to the Pfizer or Janssen. The former is more interesting, since the mechanism of action is the same for those two vaccines. 

Sputnik is mentioned twice among the other licensed vaccines, while Sinovac, another frequently used vaccine is not included. 

The biggest concern is that the authors only consider timing as a potential causative factor, while there is a complete lack of plausible mechanism. Timing does not establish causation. 

The word "imperative" is probably an overstatement. We are dealing with a few cases and over a billion vaccine doses. 

Author Response

Dear Prof. Tripp

We thank you and the reviewer for the valuable feedback and for allowing us to resubmit the manuscript for consideration for publication in Vaccines. We have carefully considered all of the comments and revised the manuscript accordingly. The suggested revisions have helped in improving this manuscript. We have addressed each of the comments below in a point-by-point response (in italics) below:

REVIEWER 3

  1. Line 52: Provide a reference for this statement. 

Response: We thank the reviewer for this suggestion. The references supporting this statement have been added to the manuscript.

  1. Line 123: I have reservations about the accuracy of diagnosis when patients self-report glaucoma. How do the investigators know these were true glaucoma cases?

Response: All the self-reported cases included in the study were diagnosed by a glaucoma specialist or an ophthalmologist. These were considered true glaucoma cases on the basis of the elevated IOP and clinical examination findings that were conveyed to the patients by their physician.

  1. Line 141: Same problem here: COVID-19 is often asymptomatic and thus, patients not reporting it does not mean much.

Response: We thank the reviewer for this suggestion. We have re-worded the sentence to “Only five patients (3.1%) had a prior history of COVID-19.” We want to bring it to the reviewer’s notice that ~80% of the cases were reported by physicians, pharmacovigilance regulators and pharmaceutical companies. Irrespective of who reported the case to VAERS, the origin of these reports are the physicians who diagnose the adverse events. We also want to reiterate, that we only included those self-reported cases where there was adequate clinical history and findings that pointed to probable diagnosis of glaucoma.

  1. Line 146: Again, patients could have had undiagnosed preexisting glaucoma. Also, with the ones having diagnosed glaucoma, the adverse event would be better described as "glaucoma exacerbation", rather than newly developed glaucoma. 

Response: We thank the reviewer for this suggestion. We have added the term “glaucoma exacerbation” to describe patients with prior history of glaucoma.

  1. Line 159 and on: All data have an unusually high SD (higher than the datum itself). This needs to be explained. 

Response: The data are heterogenous, therefore the onset interval is highly variable in the patients belonging to the same groups. Moreover, many people lack awareness about the possibility of an ocular adverse event such as glaucoma following vaccination and tend to present only when the signs are exaggerated. It is evident from the fact that more than 50% of the patients reported to their ophthalmologist only after reduced vision or complete vision loss.  However, we have now changed the data from standard deviation to standard error of mean and including median for greater clarity.

  1. Line numbers disappeared for the Discussion. 

The authors provide no possible explanation why the onset was longer after the Moderna vaccine, compared to the Pfizer or Janssen. The former is more interesting, since the mechanism of action is the same for those two vaccines. 

Response: We thank the author for their insightful feedback. We apologize for the inconvenience caused due to discontinuity of line numbers, the numbering was done on the editorial end.

We have intentionally refrained from providing any speculative pathological mechanisms associated with the vaccines. We have limited to evaluating cases that have a probable temporal relationship with vaccination and clearly outlined the need for further studies to understand the causal relationship.

  1. Sputnik is mentioned twice among the other licensed vaccines, while Sinovac, another frequently used vaccine is not included. 

Response: We thank the reviewer for pointing out the inadvertent error. It has now been fixed.

  1. The biggest concern is that the authors only consider timing as a potential causative factor, while there is a complete lack of plausible mechanism. Timing does not establish causation. 

The word "imperative" is probably an overstatement. We are dealing with a few cases and over a billion vaccine doses. 

Response: We concur with the reviewer’s viewpoint, that these data only show a temporal association between glaucoma and vaccination and no causative relationship. The manuscript clearly outlines that “The Bradford-Hill criteria include nine aspects to consider when inferring causality between events: strength of the association, consistency, specificity, temporality, bio-logical gradient, plausibility, coherence, experiment, and analogy. The current study ensures temporality, as there was a short interval of time post-vaccination and the onset of the signs of glaucoma.” We have revised the last sentence to “The current study ensures temporality, as there was a short interval of time post-vaccination and the onset of the signs of glaucoma and further studies are required to evaluate the causal relationship” in the revised manuscript.

We would like to thank the editor and the reviewer for providing us with these constructive comments and hope that with the clarifications and revisions described herein our manuscript will now be suitable for publication.

Yours sincerely,

Parul Ichhpujani on behalf of all authors

Round 2

Reviewer 3 Report

The biggest problems with this study are the number of cases, as well as the lack of any plausible mechanism. We are dealing with well over 2 billion cases, of which 161 had "glaucoma" after vaccination (some had preexisting glaucoma, which the authors still included into their statistics).

It is quite possible that we would be able to find this many cases of anything in two billion doses, including car accidents, for instance, which would be obviously unrelated, despite the timing. The authors refuse to offer any plausible mechanism, but make recommendations at the end of their conclusions. This completely lacks any scientific evidence at this point. 

The authors only address the possibility of underreporting as a potential problem with the reliability of VAERS, whereas overreporting is an equally significant, known issue. 

The conclusions strongly emphasize that there is causation, however, there is no evidence presented for that. 

The data do not follow normal distribution, and the authors did not comment on this fact. 

Author Response

Dear Prof. Tripp,

We thank you and the reviewer for their valuable feedback once again and for allowing us to resubmit the manuscript for consideration for publication in Vaccines. We have carefully considered the comments from the reviewers.

REVIEWER 3

  1. The biggest problems with this study are the number of cases, as well as the lack of any plausible mechanism. We are dealing with well over 2 billion cases, of which 161 had "glaucoma" after vaccination (some had preexisting glaucoma, which the authors still included into their statistics).
    Response: We thank the reviewer for their insightful feedback. We concur with the reviewer about their observation about the few cases. Hence, we have stated that the possibility of glaucoma as an adverse event following SARS-CoV-2 vaccines is extremely rare. The cases of the more common vaccine associated adverse events such as myocarditis and Guillain-Barré syndrome is limited to a few hundred cases as well. In this paper, we do not raise concern about the safety profile of the vaccine, on the contrary, confirm that the vaccine is safe.
  2. It is quite possible that we would be able to find this many cases of anything in two billion doses, including car accidents, for instance, which would be obviously unrelated, despite the timing. The authors refuse to offer any plausible mechanism, but make recommendations at the end of their conclusions. This completely lacks any scientific evidence at this point. 
    Response: We thank the reviewer for their incisive feedback. We reiterate the possibility of glaucoma following SARS-CoV-2 vaccination is extremely rare, however, the CDC and FDA have considered glaucoma as potential risk following vaccination and have included it in the WONDER database. Moreover, pharmacovigilance bodies have reported glaucoma cases following vaccination from all over the world. In clinical practice, we have observed that patients tend to present very late, only when there is vision impairment. This paper only focuses on evaluating a temporal relationship and the need for patients and their ophthalmologists to be vigilant in the 48 hours following vaccination.

We agree with the reviewer that there is no mechanism that establishes this association and do not want to over speculate about it either. Therefore, we have highlighted the need to evaluate this possible mechanisms that may lead to this phenomenon. 

The authors only address the possibility of underreporting as a potential problem with the reliability of VAERS, whereas overreporting is an equally significant, known issue. 
Response: We have raised the possibility of underreporting because it is commonly seen in clinical practice as well as the literature suggests that glaucoma cases are often underreported due to slow progression of the disease and the general lack of screening and awareness amongst the population. We agree that strong reliance on imaging based evaluation of a patient, Glaucoma maybe overrated in some cases (“ Red disease”). Therefore, we have changed the title of the paper to “Probable cases of glaucoma following SARS-CoV-2 vaccination”.

The conclusions strongly emphasize that there is causation, however, there is no evidence presented for that. 
Response: We have removed the sections that emphasize causation and have limited it to a temporal relationship with the SARS-CoV-2 vaccination.

The data do not follow normal distribution, and the authors did not comment on this fact. 
Response: We thank the reviewer for raising this critical issue. Since this is a study evaluating adverse event data retrospectively, it is not possible for the data to be normally distributed. The likelihood of the adverse event data in any database to be normatively is extremely low. 
